# The Effect of Dung Beetle Size on Soil Nutrient Mobilization in an Afrotropical Forest

**DOI:** 10.3390/insects12020141

**Published:** 2021-02-07

**Authors:** Roisin Stanbrook, Edwin Harris, Martin Jones, Charles Philip Wheater

**Affiliations:** 1Biology Department, University of Central Florida, Orlando, FL 32816, USA; 2Department of Agriculture and Environment, Harper Adams University, Newport TF10 8NB, UK; eharris@harper-adams.ac.uk; 3Department of Natural Sciences, Manchester Metropolitan University, Manchester M1 5GD, UK; m.jones@mmu.ac.uk (M.J.); p.wheater@mmu.ac.uk (C.P.W.)

**Keywords:** dung beetles, functional traits, nutrient transfer, soil fertility

## Abstract

**Simple Summary:**

Dung beetles are known to instigate and provide a number of services, which benefit humankind. In addition to feeding on mammal dung, dung beetles also use dung to form underground brood balls which are used for larval development. This process instigates several ecological functions, such as dung removal and nutrient cycling. Recent studies have highlighted the importance of dung removal in pastoral and natural habitats but the effect of dung beetle body size on the amount of nutrients transferred when dung is buried remains unclear. In this preliminary study, we investigate the role of African dung beetle body size in nutrient cycling. We analyzed the nutrient content of soil directly underneath dung pats using three treatments containing dung beetles of varying sizes and one control treatment without beetles over a 112-day period. Our data show that dung beetle body size has a significant effect on the quantity of macronutrients transferred over time and that dung beetle body size is an important factor in the type and amount of nutrients transferred.

**Abstract:**

Despite recognition of its importance, little is known about functional aspects of soil macrofauna. Here, we investigated the effect of dung beetle body size on macronutrient movement (N, P, K, and C) from elephant dung into soil over 112 days in an Afrotropical forest. We report a large overall effect where more macronutrients are moved into soil over time when beetles are present compared to a control treatment. We also report a large effect of beetle body size on the amount of macronutrient movement, with larger dung beetles moving more nitrogen, phosphorus, potassium, and carbon from baseline measurements when compared to smaller sized dung beetles. The presence of smaller sized dung beetles showed a significant positive effect on potassium and phosphorus transfer only. We provide the first experimental evidence that the body size of African dungs directly influences the type of macronutrients recycled and discuss the importance of dung beetle body size for maintaining soil fertility.

## 1. Introduction

Soil nutrient recycling is fundamental to the maintenance of global ecosystem services. It has been suggested that soil be viewed as natural capital that contributes to ecosystem function by maintaining the bioavailability of nutrients and physical structure of the environment [1,2], as well as contributing to human and food security [3]. There is much evidence that soil contributes to the maintenance of biodiversity and ecosystem stability, e.g., through the regulation of the microclimate and the control of pathogens [4]. However, soil arthropods defined as fauna that alter the physical structure of soil have received relatively little research attention for their role in ecosystem service provision [5]. Furthermore, the function and importance of dung in nutrient cycling is understudied despite being likely to have a critical role in soil environments. Most herbivores use only a small proportion of the nutrients they ingest; in mammals, 60–99% of the ingested nutrients are returned to the soil in the form of dung and urine [6]. Paracoprid dung beetles or “tunnelers” play an important part in removing dung from soil surfaces. Paracoprids dig tunnels below dung and bury brood balls consisting of relocated dung in nests underground. One or multiple eggs are laid in these brood balls, which then are used as support systems for developing larvae. [7]. Incidental nutrient cycling occurs when this dung is manipulated during nest building and during its subsequent and sometimes lengthy stay underground. This handling may accelerate nutrient breakdown and incorporation of macronutrients, such as fecal nitrogen, directly into the soil [8,9]. Dung beetles also aerate dung pats that changes decomposition from anaerobic to aerobic, facilitating the release of greenhouse gases [10]. This recycling of nutrients has been shown, experimentally, to increase pasture productivity through the incorporation of organic matter into substrates [11,12].

Dung beetles have been classified into functional guilds based on traits such as body size, reproductive strategy, flight activity patterns, and dung removal behavior [13,14]. There is some evidence that large paracoprid dung beetles remove greater quantities of dung from soil surfaces [15,16,17], however, the functional relationship between dung beetle trait diversity (e.g., body size or nesting behavior) and the maintenance of soil nutrient quality due to nutrient recycling remains unclear.

We investigated the effect of body size in paracoprid dung beetles on soil macronutrient recycling in an equatorial African forest ecosystem. Our aim was to test whether there is a strong functional effect of dung beetle body size on the quantity of macronutrients passed from elephant (*Loxodonta africana*) dung into the soil. Specifically, our objectives were to (1) assess whether the transfer of nutrients from dung to soils is influenced by dung beetle body size and (2) estimate the temporal effect of the dung beetles on dung to soil nutrient transfer. We discuss our findings in the context of the functional diversity of soil macrobiota and its implications for soil nutrient enrichment.

## 2. Materials and Methods

### 2.1. Study Site

The study was conducted within the Aberdare National Park (ANP), Nyeri County, Kenya (0.4167° S, 36.9500° E). The ANP is ring-fenced and is contained within the forested Aberdare range, which is an elongated mountain range, running approximately north to south, parallel to the direction of the Rift Valley, 60 km to the west of Mount Kenya [18]. The slopes are steep and densely forested while the foothills have been cleared of forest and are intensively farmed by agro-businesses who grow crops such as pyrethrum and coffee [19] but also by small holder subsistence farmers who rely on cash crops for food stability. The underlying soil is mollic Andosol [20], a part-volcanic, humus rich, and gritty loam with a high level of phosphorus absorption but low levels of phosphate availability [21].

### 2.2. Dung Beetle Classification

Dung beetles were collected using baited pitfall traps 24 h before the start of the experiment. The traps consisted of a 500 mL cup covered with an inverted funnel with a 3 cm aperture. The funnel allowed dung beetles to drop into the trap but not escape. Traps were baited by adding 50 g of elephant dung to the bottom of the cup, which attracted dung beetles into the trap. Traps were buried in the ground with the lip of the cup flush with the soil surface. All captured individuals were identified to genus. Total body length (anterior clypeal sinuation to pygidium) was measured to the nearest millimeter using digital calipers, and fresh biomass was measured to the nearest gram. Dung beetles classified based on body size were assigned to one of three size categories (see Table 1): (1) small (body size range: >5 to <15 mm), (2) medium (>15 to <25 mm), or (3) large (>25 mm). Body size classification was done using the functional classification of dung beetles proposed by Doube (1990). This system classifies dung beetles according to body size and dung exploitation behavior [13]. We also included a negative control treatment with no beetles. Each treatment type contained an equal biomass of beetles (8.1 ± 0.04 g) as this was the combined body weight of the pair of Heliocopris beetles used in the largest body size category.

### 2.3. Mesocosm Design

Two replicates of four treatment types were used to assess macronutrient transfer. The limited number of replicates was used as logistics, and sampling time was limited by the availability of Kenya Wildlife Services security personnel. Each treatment involved an experimental mesocosm containing exclosures consisting of 40 L (height: 50 cm × width 40 cm) plastic buckets buried with the top lips flush with the soil surface. Further, 40 L of excavated soil (sifted with a 2 mm aperture to remove debris and macroinvertebrates) was placed into each bucket until it was completely filled. Freshly deposited elephant dung was collected from the top section of boli, rejecting dung in contact with the ground to avoid soil contamination. Similarly, dung contaminated by urine was not used. Dung was shaped into hemispherical 1 L pats and frozen for 20 h to kill any macroinvertebrates present. Dung pats were defrosted at room temperature and one dung pat was placed on top of each soil-filled bucket and the dung beetles for each treatment type were added (Figure 1). A pyramidal structure of wooden poles wrapped in 1.2 mm gauge netting placed above each bucket prevented ingress or egress of dung beetles during the experiment (Figure 1). In the control treatment, a dung pat was placed but no beetles were added. The experiment ran for 112 days from the 28 April 2015, covering the expected completed lifecycle for all species used in the experiment, thus allowing the action of both adult and larval dung beetles to be recorded [22]. The Aberdare National Park is located almost directly on the equator, and there is little fluctuation in annual temperatures, which may have affected the dung decomposition rate.

### 2.4. Soil Samples

Soil samples were collected using a standard soil corer (2.5 × 10 cm^2^) with one core collected under each pat at the start of the experiment (day 0) and subsequently, at days 7, 14, 28, 56, and 112. Each soil sample was frozen at −20 °C prior to transport and laboratory analysis. Soil samples were dried for 24 h at 70 °C, then pulverized in a ceramic mortar to pass through a 2 mm sieve before being analyzed for nitrogen (N), phosphorus (P), and potassium (K) concentrations using the Mehlich-3 extraction procedure [23]. We added 5 g of soil to 20 mL of 0.05 M HCl in 0.025 M H_2_SO_4_, and the filtrate was analyzed for N, P, and K by inductively coupled plasma-atomic emission spectrometry (ICP-OES). Approximately 5 g of dried and weighed soil was decarbonized with 1 M solution of HCl before being analyzed for total C and N concentrations through a LECO TruSpec analyzer using the combustion (Dumas) method [24]. Data were analyzed using a linear mixed effects approach with time and functional group as explanatory variables and the amount nutrient transferred as the response variable. All analyses were completed using the nlme package [25] in R software version 3.6.1 (https://www.r-project.org/ accessed on 8 August 2019) [26].

## 3. Results

There was a highly significant effect between treatments for all macronutrients across the 112-day experimental period (all *p* < 0.05 for C, N, P, and K, see Table 2 and Figure 2). The presence of beetles significantly increased nutrient uptake in the soil for all treatments, relative to passive leaching of nutrients from dung in the absence of beetles in our control treatment. Large-bodied beetles effected the greatest change in macronutrient status, enriching the soil on average by 44.51% for all macronutrients in comparison to the control treatments.

All functional groups had a significant effect on available P transfer from dung into the soil. The available P content in each treatment increased rapidly from day 0 for all functional guilds but appeared to stabilize by day 56 of the study (Figure 2C) and then decreased. Inorganic N content in the soil from all the treatments increased rapidly from day 14 for all treatments until day 56 where they tapered off. Inorganic N content in the soil only significantly increased in the presence of the large-bodied functional guild.

When effects between treatments were analyzed, the greatest effects were observed between the control (dung + no beetles) and the functional guild containing large beetles (beetles with a body length >0.25 mm; Table 3). Thus, large-bodied beetles accounted for the greatest transfer of nutrients into the soil for all macronutrients we measured, i.e., carbon (*p* = 0.001), nitrogen (*p* = 0.002), potassium (*p* < 0.001), and phosphorus (*p* < 0.001) over time with the largest overall effect for the transfer for exchangeable potassium (Figure 2D). The small-bodied functional guild showed the smallest effect for macronutrient transfer to the control; with significant effects for K (*p* = 0.01) and P (*p* = 0.01), but not for N or C (both *p* > 0.05; see Figure 2 and Table 3). The medium-bodied functional guild showed a moderate effect on soil macronutrient enrichment with significant effects for K (*p* < 0.001), P (*p* = 0.003), and C (*p* = 0.01), but no difference for N (*p* = 0.08) (Table 3 and Figure 2B).

## 4. Discussion

The results of this preliminary study suggest that paracoprid dung beetles of all size classes have a significant positive effect on the incorporation of macronutrients from dung into the soil. When we ranked the size classes in order of their capacity to facilitate nutrient exchange, our results show that large beetles had the greatest effect, followed by medium and small-bodied beetle treatments, respectively. Our results also suggest that the movement process and rate of nutrient transfer from dung to soil differed per nutrient type. The transfer of readily available K content was much faster than those of other nutrients, irrespective of the dung beetle treatments. Most K in dung is water soluble when present as K_2_O potash compound and when the contents of water-soluble N and P in dung are relatively small [28], and the difference in movement of those nutrients from the dung to the soil are possibly explained by the difference in their water solubility.

There has been great interest in the role dung beetles may play in nutrient cycling in soils but particularly the impact they may have in agroecosystems. For example, in sandy loam soils, dung beetles have been found to increase the nutrient content of pasture soils 2–10 cm below the surface [29], resulting in the increased mineralization of organic nitrogen [8,30] and the transfer of available phosphorus, nitrogen, and exchangeable potassium by paracoprid beetles [31]. This study and Yamada et al. both used species in the genus Onthophagus to assess how dung beetle activity facilitates the transfer of nutrients into soils. Our results mostly concur with those of Yamada et al. in that the amount of available phosphorus peaks at ~30 days after dung beetles are released onto experimental pats and but then diminishes from days 28–56. However, the amount of exchangeable potassium transferred into soil rises over our study timeframe and continues to rise but peaks at day 14 and rapidly decreases until day 56 in Yamada et al. The pattern for nitrogen transfer is roughly comparable between our study and Yamada et al. and is mostly congruent with their results especially the pattern of exchange between their treatment, which contained less than 40 individuals, and our treatment using small-bodied individuals (>5 <15 mm). Both treatment types increased until day 14, fell rapidly until day 28, and then rose slightly again at day 56 (Figure 2A).

The largest beetles in our experiment are in the genus *Heliocopris*, which contained species that are among the largest dung beetles in the world [32] and are known for their ability to relocate large quantities dung underground [22,33,34]. They tend to specialize on the dung of large herbivores such as elephant and rhino and occur at relatively low population densities, most likely because of their large body size and the low density of their preferred dung type [35]. Their large body size is frequently cited as a trait that correlates significantly with increased extinction risk in both paracoprid [36,37] and telocoprid dung beetles [38], and several studies highlight declines of large-bodied dung beetles in the presence of habitat disturbance [39,40] and with declines of large herbivore density [41].

When dung beetle morphological traits were assessed in the context of ecosystem functioning, it has been reported that the absence of large, nocturnal tunnellers yielded a 75% reduction in the quantity of dung removed from soil surfaces [42]. While both large- and small-bodied dung beetles eat dung particles of the same size [43], large dung beetles tend to bury, and use, larger amounts of dung for both feeding and breeding [34]. There are various explanations for this disparity, mostly related to increased resources needed for larval survival and specialist feeding strategies [44]. Although smaller bodied species may process large amounts of dung relative to their size, our knowledge of their impact on nutrient transfer is limited. We do know that herbage yield has been found to be greater when numerous individuals of smaller bodied species are present in experimental plots compared to a fewer amount of larger bodied individuals [11] and that their presence stimulates bacterial and microbial growth in soil horizons. Other studies investigating different aspects of functional diversity have established that single species may be more influential in terms of ecosystem services provision than overall species richness [40,45]. These observations are congruent with our findings that suggest that the largest sized dung beetles are functionally the most important species in effecting soil nutrient transfer from dung in African forests, as they are more effective at burying larger quantities of dung. However, these large dung beetles, in general, appear to be the least tolerant to habitat perturbation and other drivers of ecosystem change [46,47].

Soil nutrient depletion has been linked with declines in crop productivity in sub-Saharan Africa [48], and Kenya is particularly affected by falling agricultural productivity and diminishing food security, with 12 million people residing in areas with land degradation [49]. Food webs may be linked across habitat and protected area boundaries and the biodiversity of one ecosystem, in this case, a national park, may influence the functional delivery of services to adjacent areas such as the agriculturally important land described here.

## 5. Conclusions

Despite being limited in sample size this study provides clear evidence that dung beetle body size has a significant effect on the amount of nutrients cycled into soils. Further exploration into the role in which functionally important traits affect the delivery of ecosystem services mediated by dung beetles in forests and adjacent habitats is in progress but should be broadened to encompass greater geographic [50] and spatial scales [51] as most current and historical investigations have been biased towards local studies and within limited geographic scopes.

Historically, soil fertility depletion is the major biophysical cause of declining crop productivity and a fundamental root cause for declining food security on smallholder farms in central Kenya. This may have an impact on a local scale and may indirectly affect the agriculturally dependent communities which surround the Aberdare National Park. This study reinforces the importance of understanding how functional traits of beneficial insects such as dung beetles can provide ecosystem services essential for human survival and highlights that loss of large paracoprid dung beetle species could negatively impact the transfer of important macronutrients to soil where it can be accessed by plants. It is important, therefore, to safeguard functional groups that are the most important to sustain ecosystem functioning and prioritize understanding how sensitive these species may be to anthropogenic activity.

## Figures and Tables

**Figure 1 insects-12-00141-f001:**
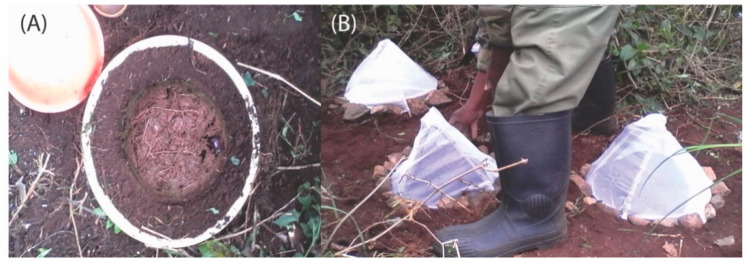
Experiment design: (**A**) a 1-L dung pat placed on top of a soil filled bucket and (**B**) pyramidal structures containing dung pats.

**Figure 2 insects-12-00141-f002:**
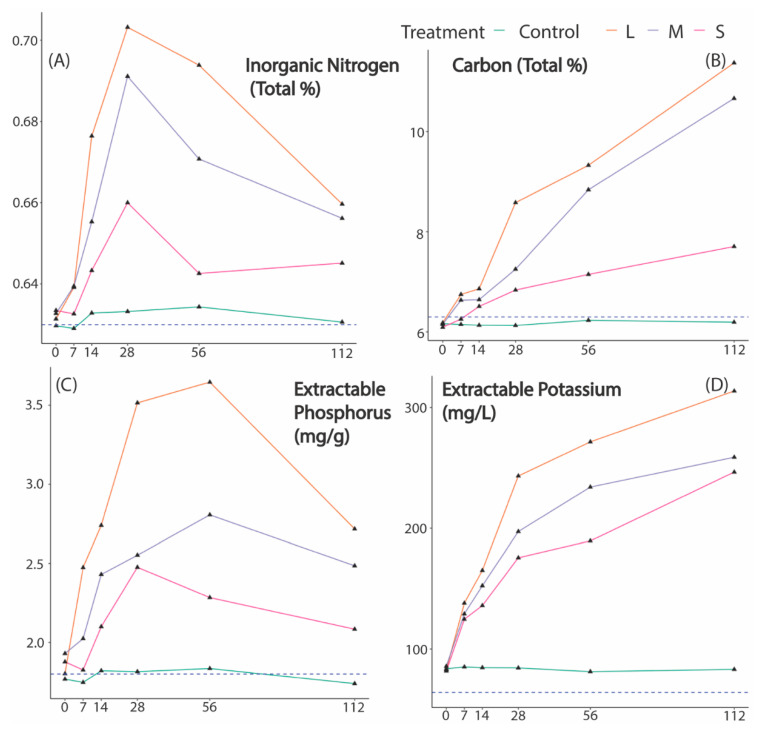
Changes in the soil nutrient contents: (**A**) total nitrogen, (**B**) total carbon, (**C**) extractable phosphorus, and (**D**) extractable potassium over 112 days. The horizontal dashed lines are published evidence of the baseline bioavailable nutrients found in Nicholson (1976) [27], a previous study in the same area and soil type as the current study.

**Table 1 insects-12-00141-t001:** Treatment, body size, and ordered proportionate composition of genera used in each replicate.

Treatment	Body Size (mm)	Genera	Number of Individuals per Genera	Proportion of Genera in Treatment (%)
Large	>25	*Heliocopris*	2	100
Medium	>15 and <25	*Onitis*	8	80
		*Diastellopalpus*	4	15
		*Copris*	6	5
Small	>5 and <15	*Onthophagus*	16	70
		*Milichus*	22	10
		*Oniticellus*	6	12
		*Liatongus*	8	6
		*Euoniticellus*	12	1
		*Caccobius*	6	1

**Table 2 insects-12-00141-t002:** Overall main effects for treatment and time (* denotes significance).

Nutrient	Treatment		Time	
	F (df)	*p*	F (df)	*p*
N	4.61 (3, 15)	0.01 *	8.31 (3, 39)	0.001 *
C	4.84 (3, 15)	0.01 *	10.16 (3, 39)	0.01 *
P	10.68 (3, 15)	<0.001 *	0.36 (3, 39)	<0.05 *
K	14.17 (3, 15)	<0.001 *	21.76 (3, 39)	<0.001 *

**Table 3 insects-12-00141-t003:** Contrast tests for levels of treatment relative to baseline (* denotes significance).

Nutrient								
Treatment Level Compared to Baseline	N		C		P		K	
	Estimate	*p*	Estimate	*p*	Estimate	*p*	Estimate	*p*
Small	0.09	0.19	1.67	0.07	0.75	0.01 *	75.33	0.01 *
Medium	0.13	0.08	2.56	0.01 *	1.21	0.003 *	92.54	<0.001 *
Large	0.26	0.002 *	3.14	<0.001 *	1.63	<0.001 *	118.72	0.001 *

## Data Availability

The data presented in this study are openly available in FigShare at https://doi.org/10.6084/m9.figshare.13713946.v1 (accessed on 11 December 2020). This manuscript contains material that has previously formed part of a PhD thesis; Stanbrook, R., The Scarabaeidae dung beetles of the Aberdare National Park, Republic of Kenya: ecosystem services and factors affecting diversity and abundance. Preprint at http://e-space.mmu.ac.uk/621286/ (accessed on 17 December 2020) and is publicly available under a Creative Commons license according to the requirements of the institution which awarded the qualification.

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
