# Peer review of "The Effect of Dung Beetle Size on Soil Nutrient Mobilization in an Afrotropical Forest"

_insects, 2021, doi:10.3390/insects12020141_

Round 1
Reviewer 1 Report
see file

Reviewer 2 Report
English grammar and style needs to be revised in many sentences. Introduction needs to be expanded to guide towards your conclusions. Methods should be clearer. Results need to be expanded and clarified. Several figures/pictures and an extended table 2 or separate table are needed providing mean values of nutrient increase/decrease with SD. Discussion should be extended. Detailed recommendations in the attached annotated manuscript PDF.

Reviewer 3 Report
This is an interesting work reporting on nutrient mobilisation from elephant dung in tropical Africa. The major drawbacks are the superficial classification of beetles (only by size, yet called by sophisticated terminology) and the low sample size. These need attention.
Specific comments (inc. a few in the body of the Ms itself, attached):
Pse consider the alternative title:
The effect of dung beetle size on soil nutrient mobilisation in an Afrotropical forest
Simple summary: you will surprise your "simple reader" to claim that dung beetles "nest" in dung. I suggest you rephrase these sentences.
Abstract: you do not indicate where the study was done, ye tit is important. Modify/revise. Additionally, your own results must be mentioned in past tense.
The last sentences of the abstract are of the 'list-type' - try to rephrase and write the results directly.
keywords: use words that do not appear as title words
L38 - what is "ecological food security"? I think there is none such. Rephrase the second part of the sentence
40-42 received little attention concerning which of their activities?
49 write a little about these "nests" - the reader will not immediately know what do you mean here? What are these "nests" for?
72-73 north to south
79 I disagree that you did any classification here by "functional guilds" - you merely classified beetles into size classes. The justification for establishing the size classes like that needs to be provided (why not 5-8mm, 8-20, etc., for example?). Why would these size classes correspond to any functional guilds?
89 how did you decide that 8.1 g of beetle biomass has to be in one treatment?
Table 1 - if you indicated the unit (mm), you do not have to repeat it in the body of the table. How did you decide to distribute the biomass within classes like that?
94 two (?) replicates? Pse explain why more was not possible to set up
135-137 contradictory statements
fig 1: move the key above the panels (currently in panel 1). Label panels by name not code - will simplify understanding. do not repeat the unit on the horizontal line - it is already on the vertical axis. Use empty circles as symbols
Table 2 - remove the multiple horizontal lines from the table. Delete "functional guild" from the headings
162 ff - you already stated this. Do not repeat.
Round 2
Reviewer 2 Report
The authors have corrected all major issues I found in my first review with a few minor issues remaining. The text should again be proofread - I caught a few new issues and marked them. I attach their latest ms with my comments as a PDF.

Author Response
Author’s response to comments
Reviewer Two:
We appreciate your comments and the thorough re-review of our manuscript ID: insects-1062968, entitled “The effect of dung beetle size on soil nutrient mobilization in an Afrotropical forest”. Below, please find our point-by-point responses (in bold) to the questions and concerns raised by Reviewer two. Line numbers in our response refer to the re-revised manuscript.
Style: Repeated use of same word
L11 The word 'incidentally' has been removed from the manuscript.
Breakup into multiple shorter sentences
L15-17 We have now broken this longer sentences into two shorter sentences to aid readability.
show that dung beetle…
L18 The word ‘that’ has now been added.
Accepted but you should state that you follow this definition of "soil fauna" in your introduction.
L41 We have now added this definition to our manuscript.
the body size of African dung beetles
L29 Corrected.
again - you did not study any functional characters besides body size so why not just label it as what it is?
L31 we have amended the text to now read ‘dung beetle body size’.
This is another term that sounds sophisticated but actually is unclear - I was not asking why time plays a role - which is obvious - but that time course is a better term than "temporal effect".
L69 We have now included the term ‘time course’ and removed the term ‘temporal effect’
My comment about possible correlations was a suggestion to try this. Wouldn't have taken much time...
As the sizes of the beetles are categorical variables (at most 3 ranks) rather than a continuous variable we agree that we could have run a Spearman's on the ranked size and nutrient content. However, as we have such clear and significant results we decided it was unnecessary as it would not have added anything further to manuscript.
double wording
L185 Extra ‘have’ removed from the sentence.
Shouldn't this be a number following Insects style?
L191 We have checked the MDPI reference and citations style guide and cannot find any information that relates to referencing in this form. Yamada et al. are referenced using the ‘Insects style’ in the previous sentence. We also checked the style guide used by the American Chemical Society as this is the style that MDPI is widely based upon. The ACS style guide uses ‘Author et al.’ for within text citations but does not italicize the latin abbreviation. With this mind we have changed the typeface for the citation and it now reads Yamada et al.’.
Reverse order
L238 We have now rephrased this